# Reproducibility of "FDA: Fourier Domain Adaptation for Semantic Segmentation" for ML Reproducibility Challenge 2020

## Reproducibility Summary

*The following paper is a reproducibility report for **FDA: Fourier Domain Adaptation for Semantic Segmentation** [12] published in the CVPR 2020 as part of the ML Reproducibility Challenge 2020. The original code was made available by the author <link>. The well-commented version of the code containing all ablation studies performed derived from the original code along with WANDB[1] integration is available at <link> with proper instructions to execute experiments in README.*

**Scope of Reproducibility**

The central claim of the paper was that the methodology did not require any training to perform domain alignment and a simple Fourier Transform could achieve state-of-the-art performance in the current benchmarks when integrated into a standard semantic segmentation model. We performed both metric and qualitative analysis of the method, comparing them with both the author's and SOTA values. Major focus was given on the GTA5 dataset specifically along with speculations on the error rate discrepancies and certain experiments to rectify the issue to an extent. Codeflows are provided for better understanding of the code-base wherever possible.

**Methodology**

We used the publicly available source code provided by the authors. Minor changes were made to the source code in order to load the model weights properly. The reproducibility experiments followed the training protocol as described in the original paper. We first tested the pre-trained models provided by the authors in the original GitHub repository. We also trained all the models mentioned in the paper from scratch and evaluated them on the given target dataset to verify the claims given in the paper.

**Results**

We verified all claims except those involving Synthia dataset. Overall, we were able to reproduce the majority of the results mentioned in the paper within 5% error using our optimized strategy compared to what was mentioned in the paper. Along with this complete review of the code-base provided by the author was done along with optimizations wherever deemed possible.

**What was easy**

The code provided in the original repository was very straight forward and well documented. The model architecture is easily implemented as it is built on a pre-existing framework (BDL)[7].

**What was difficult**

The significant challenges faced in reproducing the results in the chosen publication were computation bound in nature, concerning mainly the batch size, long training hours (40-60 hours) and large CPU RAM for pseudo label generation.

The huge training time along with the vast number of models to be trained became a major bottleneck for conducting even rudimentary ablation studies like hyperparameter search.

**Communication with original authors**

Contact was made with the authors via email regarding the computational requirements of the training and errors in training to which prompt and helpful replies were given by them.

# 1 Introduction

Unsupervised domain adaptation (UDA) refers to adapting a model trained with annotated samples from one distribution (source), to operate on a different (target) distribution for which no annotations are given. Simply training the model on the source data does not yield satisfactory performance on the target data due to the covariate shift. State-of-the-art UDA methods require difficult adversarial training but the method given in the paper computed the (Fast) Fourier Transform (FFT) of each input image, replacing the low-level frequencies of the target images into the source images before reconstituting the image for training, via inverse transform (iFFT), using the original annotations in the source domain. As a paragon, state-of-the-art model with adversarial training was used [6]. A variety of sizes as well as a multi-scale method consisting of averaging the results arising from different domain sizes were tested in the paper.

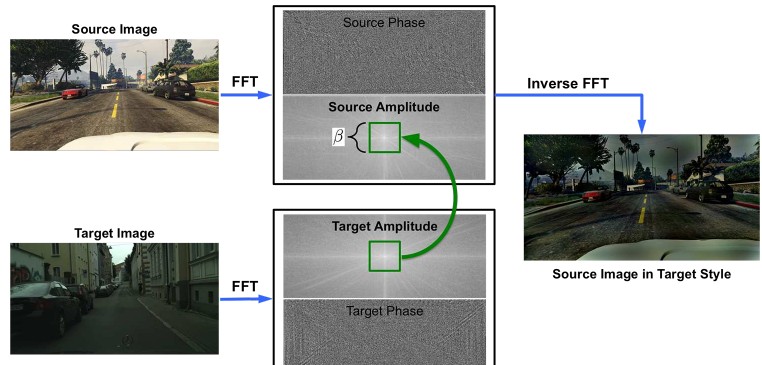

Figure 1: **Spectral Transfer:** Mapping a source image to a target "style" without altering semantic content

# 2 Scope of reproducibility

The paper revolves around the claim that state of the art performance in unsupervised domain adaptation can be achieved using the simple method of Fourier transformation, which does not require any extra training, eliminating the need of difficult adversarial methods. In the paper, models are trained on different values of $\beta$ and self-supervised training is also implemented. Models are trained on 2 backbones - ResNet101 and VGG16 and results are obtained by doing domain adaptation from GTA5-> CityScapes and Synthia-> CityScapes.

Hence the claims can be summarized as follows:-

**GTA5->Cityscapes**:

1. The fully trained MBT ResNet101 model achieved 50.45% mIoU when trained using FDA, surpassing the previous state-of-the-art method (BDL) by 1.95%.
2. The fully trained MBT VGG model achieved 42.2% mIoU when trained using FDA, surpassing the previous state-of-the-art method (BDL) by 0.9%.

**Synthia->Cityscapes**

1. The fully trained MBT ResNet101 model achieved 52.5% mIoU when trained using FDA, surpassing the previous state-of-the-art method (BDL) by 1.1%.
2. The fully trained MBT VGG16 model achieved 40.5% mIoU when trained using FDA, surpassing the previous state-of-the-art method (BDL) by 0.5%.

## 3 Methodology

The authors have made the source code associated with the paper publicly available on GitHub, as well as links to download their pretrained ResNet-based and VGG-based models. To train the models, we followed the training protocol, as described in the original paper. The ResNet101 and VGG16 models, which were pretrained as described in [7], were employed as the backbone of the model. The codeflow is described in Figures 3 and 6.

### 3.1 Method descriptions

In unsupervised domain adaptation, we are given a source dataset $D^s = \{(x_i^s, y_i^s) \sim P(x^s, y^s)\}_{i=1}^{N_s}$, where $x^s \in \mathbb{R}^{H \times W \times 3}$ is a color image, and $y^s \in \mathbb{R}^{H \times W}$ is the semantic map associated with $x^s$. Similarly $D^t = \{x_i^t\}_{i=1}^{N_t}$ is the target dataset, where the ground truth semantic labels are absent. Generally, the segmentation network trained on $D^s$ will have a performance drop when tested on $D^t$. The proposed Fourier Domain Adaptation (FDA) technique reduces this gap between the source and target domains. As described in Eq 1, the Fourier Transform, $F$ for a RGB image and can be calculated and efficiently implemented as described in [4]. Accordingly, $F^{-1}$ is the inverse Fourier transform that maps spectral signals in the Fourier domain back to the image space.

$$\mathcal{F}(x)(m,n) = \sum_{h,w} x(h,w) e^{-j2\pi\left(\frac{h}{H}m + \frac{w}{W}n\right)}, j^2 = -1 \tag{1}$$

The method requires selecting the size of the spectral neighborhood to be swapped (signified by a green square in Figure 1). We define a mask $M_\beta$ wherein all values are zero except for the centre region where $\beta \in (0,1)$:

$$M_\beta(h,w) = 1_{(h,w) \in [-\beta H:\beta H, -\beta W:\beta W]} \tag{2}$$

Given two randomly sampled images $x^s \ D^s$, $x^t \ D^t$, Fourier Domain Adaptation can be formalized as described in Eq. 3, where the low frequency part of the amplitude of the source images $F^A(x^s)$ is replaced by that of the target image $x^t$.

$$x^{s\to t} = \mathcal{F}^{-1}\left(\left[M_\beta \circ \mathcal{F}^A\left(x^t\right) + (1 - M_\beta) \circ \mathcal{F}^A\left(x^s\right), \mathcal{F}^P\left(x^s\right)\right]\right) \tag{3}$$

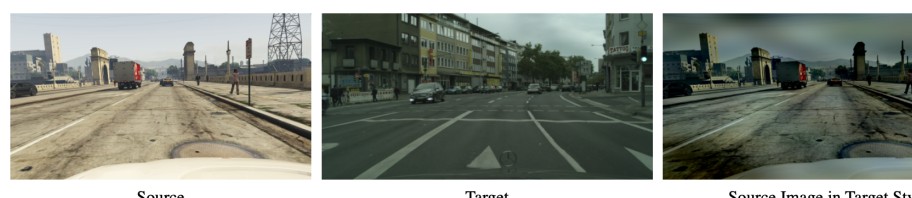

Source      Target      Source Image in Target Style

Figure 2: An example of FDA for domain adaptation for $\beta = 0.01$

Since our training entails different values of $\beta$ in the FDA operations, a self-supervised training using the mean prediction of different segmentation networks **Multi-band Transfer (MBT)** was employed. This generally lead to an increase of mIoU from its constituent models. We instantiate $M = 3$ segmentation networks $\phi_{\beta_m}^w, m = 1, 2, 3$ which are all trained from scratch and the mean prediction for a certain target image $x_i^t$ can be obtained by Eq 4.

$$\hat{y}_i^t = \arg\max_k \frac{1}{M} \sum_m \phi_{\beta_m}^w\left(x_i^t\right) \tag{4}$$

The exact methodology has been documented and a code flow for the same has been provided in Figure 3.

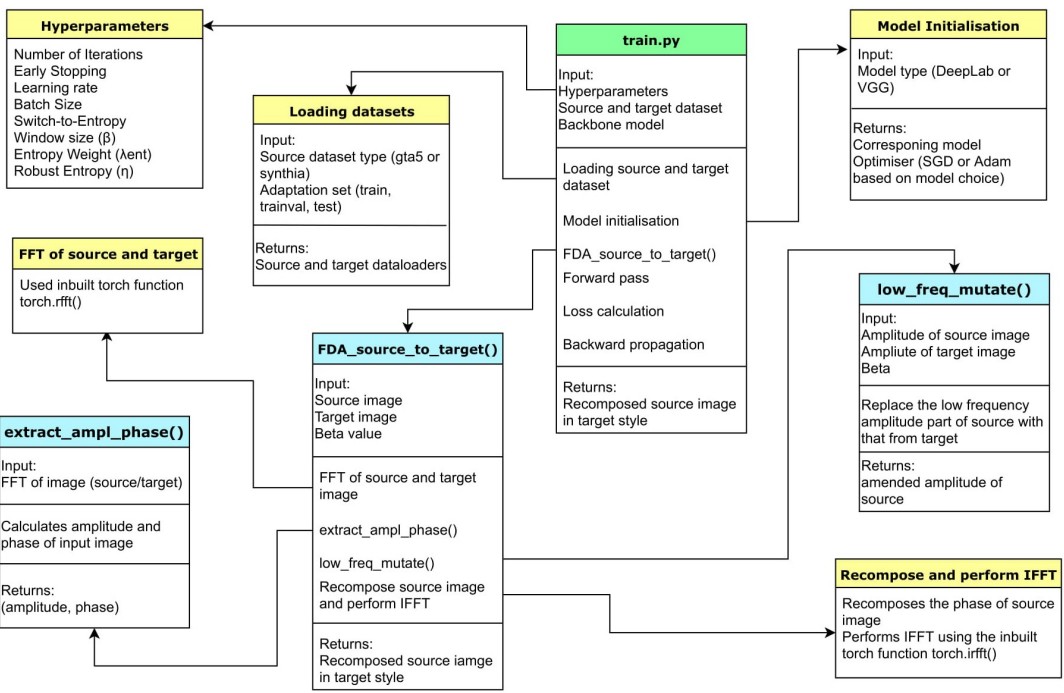

Figure 3: Method code flow

## 3.2 Datasets

GTA5 [9] and Synthia [10] datasets were used as source domain datasets and CityScapes[3] as the target domain dataset. All of the datasets were open-sourced. All the images were resized to 1280x720 and then randomly cropped to 1024x512. Further details can be found in Table 1.

| Dataset | Number of Images |
|---|---|
| GTA5 | 24966 |
| Synthia | 9400 |
| Cityscapes (train) | 2975 |
| Cityscapes (val) | 500 |

Table 1: Datasets

## 3.3 Hyperparameters

Default values of the hyperparameters given in the paper were taken and changes were made to $\beta$ and $\lambda_{ent}$ values only. The hyperparameters are listed in Table 2.

## 3.4 Experimental setup

The training code was run on Google Colaboratory with GPU (NVIDIA-SMI 460.27.04, Driver Version: 418.67, CUDA Version: 10.1). Initially, the pseudo labels were being generated on a virtual machine on the Google Cloud Platform with 40 GB memory and Nvidia Tesla T4 GPU(NVIDIA-SMI 450.51.06 Driver Version:450.51.06 CUDA Version: 11.0 ) which was later optimised, making it executable on Google Colaboratory henceforth.

| Hyperparameter | Value |
|---|---|
| Number of Iterations | 150000 |
| Early Stopping | 100000 |
| Learning rate | 2.5e-4 |
| Momentum | 0.9 |
| Weight decay | 0.0005 |
| Power | 0.9 |
| Batch Size | 1 |
| Switch-to-Entropy | 50000 |
| Window size ($\beta$) | 0.01, 0.05, 0.09 |
| Entropy Weight ($\lambda_{ent}$) | 0.005 |
| Robust Entropy ($\eta$) | 2 |

Table 2: Hyperparameters

## 3.5 Computational requirements

The ResNet101-based and VGG16-based models had different requirements on the computational expectations. The ResNet101 models took around 60 hours for training from scratch, while the VGG16 models took around 40 hours for training from scratch. The evaluation script took around 30 minutes for complete execution.

## 4 Results

The following experiments/ablation studies support the claims made earlier. We verified the claims made in the paper, which involved the GTA5 dataset, while the claims regarding Synthia dataset were untested. The detailed description of the experiments and their results to support the claim are listed below:-

## 4.1 Experiments on GTA5 data set

### 4.1.1 Baseline model on DeepLabV2

DeepLabV2[2] with ResNet101[5] was trained using SGD along with 'poly' learning rate scheduler and weight decay. Various models were trained for three different $\beta$ values and self-supervised training was performed at each round. Results of the training are given in Table 3.

| Experiment | mIoU (Paper) | mIoU (Final Checkpoint) | Error ( %) |
|---|---|---|---|
| 0.01(T=0) | 44.61 | 42.71 | 4.25% |
| 0.05(T=0) | 44.6 | 40.98 | 8.11% |
| 0.09(T=0) | 45.01 | 41.35 | 1.33% |
| 0.09 ($\lambda_{ent} = 0$) | 44.64 | 42.62 | 4.52% |
| 0.09 (SST) | 45.42 | 41.82 | 7.92% |
| MBT (T=0) | 46.77 | 44.41 | 5.04% |
| 0.01(T=1) | 47.03 | 45.02 | 4.27% |
| 0.05(T=1) | 46.8 | 45.13 | 3.56% |
| 0.09(T=1) | 46.71 | 45.03 | 3.59% |
| MBT(T=1) | 48.14 | 45.38 | 5.73% |
| 0.01(T=2) | 48.77 | 45.90 | 5.88% |
| 0.05(T=2) | 47.86 | 45.34 | 5.26% |
| 0.09(T=2) | 47.03 | 43.61 | 7.27% |
| MBT(T=2) | 50.45 | 46.14 | 8.54% |

Table 3: Results of our training of DeepLab model on GTA5 dataset

### 4.1.2 Baseline model on VGG16

FCN-8s[8] with VGG16[11] backbone was trained using Adam optimizer with a learning rate of 1e-5 which decreased by a factor of 0.1 every 50000 steps until 150000 steps. Various models were trained for three different $\beta$ values and self-supervised training was performed at each round. For the VGG16 model, only the final MBT mIoU was mentioned in the paper. We observed an improvement in mIoU for all the 3 models after each round. However, in the last round, we did not observe any improvement in the MBT mIoU. We present the results of all the rounds of our training below. Results of our training are mentioned in Table 4.

| Experiment | mIoU (best) |
|---|---|
| 0.01 (T=0) | 35.57 |
| 0.05 (T=0) | 34.77 |
| 0.09 (T=0) | 34.13 |
| MBT (T=0) | 36.48 |
| 0.01 (T=1) | 39.23 |
| 0.05 (T=1) | 39.61 |
| 0.09 (T=1) | 38.89 |
| MBT (T=1) | 40.08 |
| 0.01 (T=2) | 39.52 |
| 0.05 (T=2) | 40.96 |
| 0.09 (T=2) | 39.42 |
| MBT (T=2) | 40.06 |

Table 4: Results of VGG model trained on GTA5 dataset

Comparisons of the final MBT mIoUs for DeepLab and VGG models trained on the GTA5 dataset can be found in Table 5.

| Experiment | mIoU (paper) | mIoU ( Final checkpoint) | mIoU ( best checkpoint) |
|---|---|---|---|
| DeepLab-MBT (T=2) | 50.45 | 46.14 | 47.42 |
| VGG-MBT (T=2) | 42.2 | 39.42 | 40.06 |

Table 5: Results of final MBT mIoUs for DeepLab and VGG.

### 4.2 Experiments on Synthia data set

Training on the Synthia dataset couldn't be performed due to unresolved issues. The Synthia dataset had to be trained on 16 classes, unlike the earlier GTA5 source dataset, which already had 19 classes. Mapping of the labels had to me done to match with the classes present in the Cityscapes dataset. The issue was later resolved as we had communication with the authors. But due to the computational and time constraints, the training of the models had to be dropped. The data-loader for the training on Synthia dataset could not be loaded with the pretrained weights. This issue couldn't be resolved which along with the aforementioned constraints led to dropping of the training.

### 4.3 Qualitative Results

Visual comparison of the model results with the author's results and the ground truth labels can be found in Figure 4. As can be observed, the predictions from our model don't appear to be noisy, as observed by the authors in their publication. Our model also maintained fine structures like poles which can be observed more prominently in the third and fifth row. Moreover the model showed greater performance on rare classes like the truck in the last row, and the bicycle in the third row. We accredit this to both the generalizability of the single scale FDA, and the regularized SST by the Multiband Transfer. Thus, we were able to verify the qualitative claims made by the author in their original publication. Beside this, the results obtained were better than author's for some classes like sky which can be backed by the class IoU results in Table 6.

| Classes | DeepLabV2(Author) | DeepLabV2 (Ours) | VGG-16 (Author) | VGG-16(Ours) |
|---|---|---|---|---|
| road | **92.55** | 90.56 | 86.12 | **88.12** |
| sidewalk | **53.34** | 44.31 | 35.05 | **41.16** |
| building | 82.36 | **82.97** | **80.61** | 80.38 |
| wall | **26.53** | 23.69 | **30.76** | 29.86 |
| fence | 27.6 | **31.89** | 20.43 | **22.55** |
| pole | **36.44** | 34.17 | 27.5 | **27.98** |
| light | **40.58** | 36.32 | **30.02** | 29.51 |
| sign | **38.87** | 30.44 | **26.01** | 22.41 |
| vegetation | 82.27 | **84.68** | 82.13 | **82.49** |
| terrain | 39.83 | **42.07** | 30.26 | **32.69** |
| sky | 78 | **79.15** | **73.63** | 72.74 |
| person | **62.6** | 61.39 | 52.52 | **52.55** |
| rider | **34.4** | 27.18 | 21.66 | **23.63** |
| car | **84.91** | 82.21 | **81.65** | 81.5 |
| truck | 34.13 | **38.04** | **23.97** | 23.34 |
| bus | **53.12** | 52.02 | **30.5** | 22 |
| train | **16.87** | 0.12 | **29.85** | 1.46 |
| motocycle | 27.7 | **29.49** | **14.58** | 10.61 |
| bicycle | **46.42** | 40.66 | **24.02** | 16.19 |
| mIoU | **50.45** | 47.97 | **42.17** | 40.06 |

Table 6: Class-wise mIoU comparison of our models with author's models

## 4.4 Effect of $\beta$

It was observed that the performance of the model was indiscriminate to the choice of $\beta$. The mIoU difference between the models for different values of $\beta$ was not more than 1.5%, thus establishing the robustness of the proposed domain adaptation technique.

We also verify the author's observation that increase in $\beta$ introduces the artifacts in the image when we swap the spectrum as evident in the Figure 5.

## 4.5 Improvements (Extra Experiments)

### 4.5.1 Using best checkpoints

Two approaches were employed during model training. In the first approach, the final checkpoints of the previous round were taken to generate pseudo labels for subsequent rounds. In the second approach, pseudo labels were generated using the best performing (in terms of mIoU) checkpoints of the previous rounds. It was found that the latter approach produced results that were closer to the paper's original results and even surpassed them in some cases. Author's approach on pseudo label generation in the paper but could be found in the official GitHub repository. We show the comparisons of the best checkpoints with paper and final checkpoints below in Table 7.

### 4.5.2 Improved the Pseudo label generation code

The pseudo label generation code given in the original repository had extensive hardware requirements, and the whole process of generating pseudo labels took around 35-40 GB of CPU RAM. Optimization of the process led it to be executable on Google Colaboratory with a less than 15 GB memory requirement. The code flow for the optimized pseudo label generation has been outlined in Figure 6. Optimization was done via saving the compressed numpy arrays as cache memory rather than in the processing memory itself. Around 5.5 GB cache memory was required to save the files, and they were deleted after the code was run completely, thus not taking any extra disk space.

No discernible difference could be found between the pseudo labels from the original code and the optimized code's pseudo labels as can be observed from Figure 7 as proof of correctness of the new code.

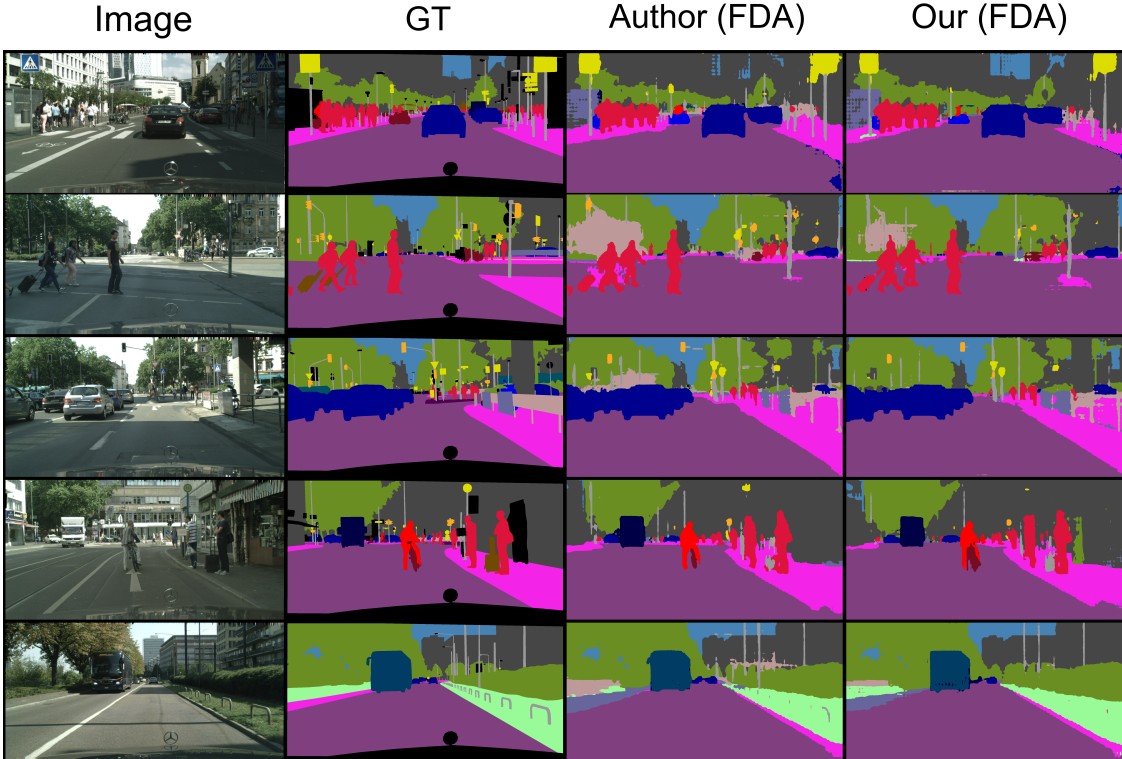

Figure 4: Visual Comparison. Left to right: Input image from CityScapes, ground-truth semantic segmentation, Author's results from DeepLab FDA-MBT, Our best FDA-MBT. Note that the predictions from our model and author's model are generally similar in terms of capturing rare classes and fine features. Moreover, our model performs better on some classes like sky as seen in the first row.

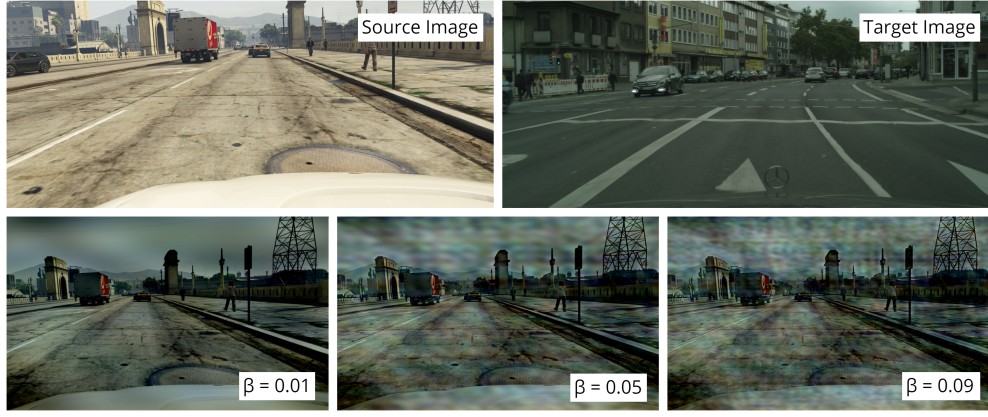

Figure 5: Effect of the size of domain $\beta$: increasing $\beta$ will decrease the domain gap but introduce artifacts

### 4.5.3   Added automatic saving of Adam optimizer

During the training of VGG models, it was found that the mIoU of the checkpoints dipped drastically when the training was resumed after a halt from unexpected reason. We later found that the intermediate optimizer states were not being saved which is essential to some extent, especially for Adam optimizer. Hence, we modified the code to save the state dictionary of the optimizer after every 2500 steps. This might have lead to an improvement in our results and made the training process independent of any interruptions.

| Experiment | mIoU (paper) | mIoU (ours) | mIoU (best) | Error |
|---|---|---|---|---|
| 0.01 (T=0) | 44.61 | 42.71 | 44.36 | 0.56% |
| 0.05 (T=0) | 44.6 | 40.98 | 41.79 | 6.30% |
| 0.09 (T=0) | 45.01 | 41.35 | 42.84 | 4.82% |
| 0.09 ($\lambda_{ent} = 0$) | 44.64 | 42.62 | 42.62 | 4.52% |
| 0.09 (SST) | 45.42 | 41.82 | 42.54 | 6.34% |
| MBT (T=0) | 46.77 | 44.41 | 45.89 | 1.88% |
| 0.01 (T=1) | 47.03 | 45.02 | 47.37 | -0.72% |
| 0.05 (T=1) | 46.8 | 45.13 | 45.56 | 2.64% |
| 0.09 (T=1) | 46.71 | 45.03 | 44.88 | 3.91% |
| MBT (T=1) | 48.14 | 45.38 | 47.97 | 0.35% |
| 0.01 (T=2) | 48.77 | 45.9 | 46.65 | 4.34% |
| 0.05 (T=2) | 47.86 | 45.34 | 45.85 | 4.19% |
| 0.09 (T=2) | 47.03 | 43.61 | 45.26 | 3.76% |
| MBT (T=2) | 50.45 | 46.14 | 47.42 | 6.00% |

Table 7: Results of DeepLab model evaluated using best checkpoints.

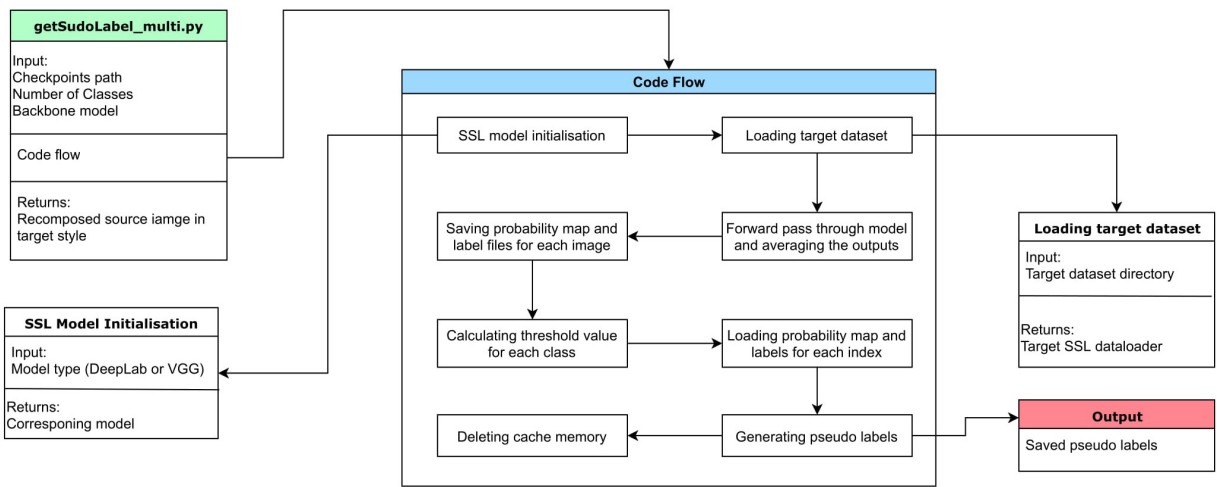

Figure 6: Optimised Pseudo Label Generation Pipeline

# 5  Discussion

Reproducibility of the results posed several obstructions, especially towards students with limited/no access to server-grade computations. In order to further study the extent of improvements made by the methods, some other experiments were carried out. We followed the two approaches, as mentioned earlier for training. The results produced by using final checkpoints were under 9% of the reported values. While the results produced by using the best checkpoints during pseudo label generation improved upon our former approach and were under 6.5% of the reported values in the paper. This can be attributed to various facts like random cropping of the dataset images for training which is bound to produce differences since no mention of the seed used by the authors could be found in both the paper and the code repository. Along with this frequent interruptions to training resulting in the loss of the optimizer weights created an impact on the results. However, in a few cases, our later approach surpassed the values reported in the paper. From our results, we can conclude that the use of Fourier Transformation does improve the results without any extra training required for unsupervised domain adaptation and add robustness to the domain adaptation problem.

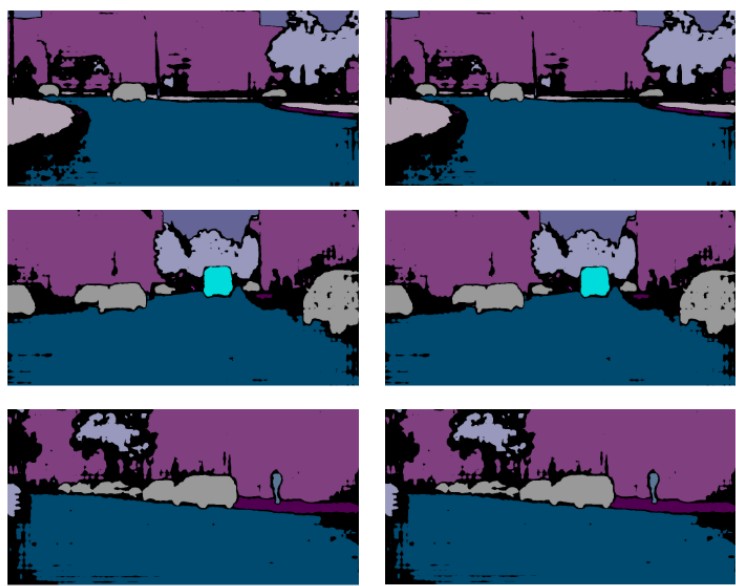

Figure 7: There is no discernible difference between the pseudo labels from the original code (Left) and the ones from our optimised code (Right)

### 5.1 What was easy

The pretrained models and original source code had been shared by authors. The code provided in the original repository was very straight forward and well documented. One can try out the models with little effort as the implementation of the method was relatively easy and a standard machine learning framework, PyTorch, was used.

### 5.2 What was difficult

Initially, the computation was started in Google Cloud Platform on a Tesla T4 GPU. A more cost-effective approach was found in using Google Colaboratory along with a premium Google Drive account for storage of checkpoints and labels. The generation of pseudo-labels was computationally very heavy, requiring around 35-40GB of memory for execution. However, this was improved through optimization of pseudo label generation, making it possible to execute on Google Colaboratory.

### 5.3 Communication with Authors

Contact with the authors was conducted via email and was restricted to the above-mentioned memory issue while generating the pseudo labels, hardware requirements and issues encountered while training on Synthia dataset due to a mismatch in the number of classes.

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
