# OpenReview forum: "[RE] FDA: Fourier Domain Adaptation for Semantic Segmentation"
_ML_Reproducibility_Challenge/2020 — Reject_

### Official Review · AnonReviewer2 · 2021-02-28
**Reproduced results without thorough hyperparameter search**

**Rating:** 7
**Confidence:** 4

**Review:**

The proposed paper reproduces the result of "Fourier Domain Adaptation for Semantic Segmentation (CVPR 2020)", by primarily utilizing the code released by the original authors. The paper has clearly outlined the details of the algorithm and also compared the results with those presented by the original authors. However, the following points might improve the quality of the work

1) There seems to be a difference between the result obtained in the paper with that presented by the original author. A detailed discussion of the possible reason for this difference is expected.

2) Although, we understand that it might be difficult to secure access to computing resources. Nevertheless, it is expected to perform a thorough hyperparameter search.

**Familiar With The Original Paper:**

I have read the original paper

**Reproducibility Summary:**

Report has summary

---

### Official Review · AnonReviewer3 · 2021-03-02
**Well structured**

**Rating:** 6
**Confidence:** 3

**Review:**

The paper is well structured and easy to follow. The authors provide a good summary with well defined scope of reproducibility. However, the authors could not verify claims on Synthia dataset but instead try to debrief the experimental setup and overall method.

Overall, the authors are able to evaluate approach on GTA5 dataset along with providing computation details for training time.  The authors perform experiments with multiple backbone and are able to reproduce within decent error rate. The limitation is authors do not show any qualitative evaluation for comparison with paper which is crucial for semantic segmentation analysis. Effect of the size of the domain β is not analyzed qualitatively. Authors provide detailed experimental setup along with summary of code flow which helps in navigating the source code.  The authors had communicated with original authors to resolve queries. They do not provide any additional/extra experiments

It is not clear in sec 4.3.1 authors mention 'we could not find the author’s approach to generate pseudo labels in the paper' but in sec 4.3.2 they claim to improve the original pseudo label approach ? Section 4.3.3 may not be relevant here.



**Familiar With The Original Paper:**

I have not read the original paper

**Reproducibility Summary:**

Report has summary

---

### Official Review · AnonReviewer1 · 2021-03-02
**Reproducibility report for paper - FDA: Fourier Domain Adaptation for Semantic Segmentation**

**Rating:** 5
**Confidence:** 3

**Review:**

The authors have provided a summary of their experiments regarding the reproducibility of the paper - FDA: Fourier Domain Adaptation for Semantic Segmentation which was published in the CVPR 2020. The central claim of the original paper is that a simple Fourier transform can be used to achieve state of the art performance when tested on the semantic segmentation task. The authors of this reproducibility report were able to reproduce most of the results from the original paper (except on one dataset) and they optimized the original paper’s code to enable an easier loading of the model weights.

Pros:
Including a codeflow in the paper is very nice. It tries to show how the overall structure of the codebase is to make it easier for anyone trying to implement and reuse code.

Typos et al for improvements:
The authors can definitely benefit from cleaning their document and making sure tables and references are hyperlinked appropriately. Some of these typos and pointers are mentioned below:

- Line 43: (include a space at new line start) … methods. In the …
- Hyperlinks in some equations are missing - for example, in Eq1 (line 64) and Eq 3 (line 68-69) . Authors might include those as it helps in easier readability and navigation.
- Some of the citations are missing / wrong. For example, citation numbered [13] in Line 65 does not lead to anywhere. There are only 10 citations under references.
- Hyperlinks for tables - for example, for Table 3, 4, 5, 6
- Table 1 and 2 are not linked anywhere (though they appear in the vicinity of the related discussion)
- Typo: varified -> verified (line 88)
- table 6 -> Table 6 (line 120)
- However in the last round -> However in the last round (line 100)
- MBT mIoU. . -> MBT mIoU. (Line 101)
- training of the mordels -> training of the models (line 110)
- via mail -> via email (line 157) —> Unless this was really done via snail mail

The claims listed under the "Scope of reproducibility" section can be improved to reflect what the authors of the reproducibility report did and what to expect in the rest of the document.

The "Method descriptions" section can be improved. The equations and used notation is not very clear. If this section can provide a good overview of the methods which is self-sufficient to understand, it will be helpful in better readability.

The authors can add some additional ablation studies to dig into and understand better the ideas from the original paper.

Overall, by addressing these issues and cleaning the document, the authors can improve their writing significantly and can be taken into consideration while making the decision.


**Familiar With The Original Paper:**

I have read the original paper

**Reproducibility Summary:**

Report has summary

---

### Decision · Program_Chairs · 2021-03-31

**Decision:**

Reject

**Comment:**

Overall reviews and/or the paper content not good enough for the AC to recommend to the journal.